# Canthin-6-Ones: Potential Drugs for Chronic Inflammatory Diseases by Targeting Multiple Inflammatory Mediators

**DOI:** 10.3390/molecules28083381

**Published:** 2023-04-11

**Authors:** Zongying Zhang, Anqi Wang, Yunhan Wang, Weichen Sun, Xiaorong Zhou, Qiuyun Xu, Liming Mao, Jie Zhang

**Affiliations:** 1Department of Immunology, School of Medicine, Nantong University, 19 Qixiu Road, Nantong 226001, China; 2Basic Medical Research Center, School of Medicine, Nantong University, Nantong 226019, China

**Keywords:** Canthin-6-one, chronic inflammatory disease, inflammasome, NLRP3, NO, NF-κB

## Abstract

Chronic inflammatory disease (CID) is a category of medical conditions that causes recurrent inflammatory attacks in multiple tissues. The occurrence of CID is related to inappropriate immune responses to normal tissue substances and invading microbes due to many factors, such as defects in the immune system and imbalanced regulation of commensal microbes. Thus, effectively keeping the immune-associated cells and their products in check and inhibiting aberrant activation of the immune system is a key strategy for the management of CID. Canthin-6-ones are a subclass of β-carboline alkaloids isolated from a wide range of species. Several emerging studies based on in vitro and in vivo experiments reveal that canthin-6-ones may have potential therapeutic effects on many inflammatory diseases. However, no study has yet summarized the anti-inflammatory functions and the underlying mechanisms of this class of compounds. This review provides an overview of these studies, focusing on the disease entities and the inflammatory mediators that have been shown to be affected by canthin-6-ones. In particular, the major signaling pathways affected by canthin-6-ones, such as the NLR family pyrin domain containing 3 (NLRP3) inflammasome and the NF-κB signaling pathway, and their roles in several CIDs are discussed. Moreover, we discuss the limitations in studies of canthin-6-ones and provide possible solutions. In addition, a perspective that may suggest possible future research directions is provided. This work may be helpful for further mechanistic studies and possible therapeutic applications of canthin-6-ones in the treatment of CID.

## 1. Introduction

Chronic inflammatory disease (CID) is a term to describe disease entities characterized by persistent inflammation. The development of this class of diseases is associated with aberrant immune responses to normal tissue substances or particular offending agents due to the occurrence of abnormalities in the immune system [1,2]. In response to injuries or some offending agents, the immune system may send out inflammatory cells and mediators to start the healing process or to eliminate infections. After this process, the inflammatory factors can be downregulated by several mechanisms. Failure of downregulation may result in persistent stimulation of inflammatory factors to the host tissues. The development of CID is usually due to persistent exposure to toxins, untreated acute inflammation, and autoimmune disorders, and may be associated with a wide range of conditions such as type 2 diabetes [3], rheumatoid arthritis [4], Alzheimer’s disease [5], inflammatory bowel disease [6], and cancer [7]. Therefore, unchecked chronic inflammation may play a detrimental role to the host and mediate a variety of human diseases [8]. Appropriate management of the duration and strength of inflammation is an effective approach by the host immune system to maintain homeostasis in the body.

The progression of modern pharmaceutical technologies has allowed the development of thousands of chemical drugs, which have largely facilitated the treatment of CID. A considerable number of these drugs are extracted from traditional herbal medicines, which have been recognized as complementary or alternative medicines in the West [9]. Canthin-6-one (1) is a β-carboline alkaloid initially extracted from the Australian plant *Pentaceras australis*, an Australian rainforest tree in the citrus family with multiple biological activities [10]. Several recent studies revealed the role of canthin-6-one (1) and its derivatives in suppressing inflammatory responses in several human diseases [11,12,13,14,15,16,17], such as inflammatory bowel disease. To date, no study has summarized the current findings of these studies in the literature. This review discusses the effects and underlying mechanisms of canthin-6-ones on several entities of CID, as well as the affected pathways and inflammatory mediators. This study may contribute to further investigation toward understanding the regulatory mechanisms of canthin-6-ones in CID and may be of benefit to the development of anti-inflammatory drugs using canthin-6-ones and related chemicals.

## 2. Unchecked Inflammation and CID

The inflammatory response is tightly regulated in a highly coordinated network consisting of many cell types from both the innate and adaptive immune system, which mediate local responses to tissue damage and infections [18]. The innate immune system relies on pattern recognition receptors (PRRs) to sense pathogenic microorganisms and other endogenous or exogenous pathogens. Through activation of PRRs, mainly expressed in innate immune cells (such as macrophages, neutrophils, and dendritic cells) [19], pathogen-associated molecular patterns (PAMPs) can trigger the inflammatory response of innate immune cells. For example, as the central part of the innate immune system, monocytes/macrophages act as sentinel coordinators of immune activity and homeostasis [20,21]. Activated macrophages produce a variety of pro-inflammatory mediators such as Nitric oxide (NO), Prostaglandin E2 (PGE2), and cytokines. Moreover, tissue-resident mast cells release many other types of mediators, including cytokines, chemokines, histamine, and prostaglandins [22]. These cells and inflammatory mediators facilitate the infiltration of other cell types, such as lymphocytes, to amplify the inflammation [12,13,14]. Some PRRs are also able to sense various endogenous signals that arise during tissue or cell damage and are commonly referred to as danger-associated molecular patterns (DAMPs) [23]. For instance, upon tissue injury, the tissue-resident macrophages and γδ T cells sense the damaged tissue via PRRs and release mediators to recruit other immune cells. Neutrophils are the first type of circulating cells attracted to the injured site, which induce the recruitment of monocytes/macrophages; the latter then sense DAMPs released from damaged tissues and thus produce mediators for tissue repair.

The process of inflammation involves several microcirculatory events, including changes in vascular permeability, leukocyte recruitment, and accumulation [24], which synchronize with the activation of various signaling pathways in resident tissue cells and infiltrating immune cells from the blood, such as those mediated by the NLRP3 inflammasome, nuclear factor-κB (NF-κB), the Janus kinase/signal transduction and activator of transcription (JAK/STAT), Mitogen-Activated Protein Kinases (MAPKs), and PI3K-AKT pathways [25,26,27,28,29]. The activation of these signaling pathways promotes the transmission of particular signaling molecules which then trigger the production of many pro-inflammatory cytokines and mediators. The latter participate in the clearance of infectious agents or act as mediators to recruit other inflammatory cells. They may also serve as potential biomarkers for disease diagnosis, prognosis, and therapeutic decisions [30,31].

As inflammation is an event that aims to remove infectious agents and initiate tissue repair, it is thus designed to be a finite process that may resolve after infections are removed or the tissue repair is complete. The resolution of inflammation involves programmed cell death and the clearance of activated inflammatory cells [32,33], and it is likely an active process, since recent evidence showed that during the resolution of inflammation, many mediators are actively generated by the host to promote the return to homeostasis [34]. For instance, several families of lipid mediators play an important role in blocking the production of pro-inflammatory cytokines and leukocytic infiltration as well as promoting the uptake of apoptotic cells [35]. The resolution of inflammation is tightly regulated by both stromal cells and immune cells. Failure of this process may cause many diseases. A strong piece of evidence supporting this idea is provided by the function of neutrophils during the inflammatory response. Neutrophils act as the first line of defense and initiate an acute inflammatory response which engulfs dead cells and tissue debris to facilitate tissue repair. However, excessive activation and persistent infiltration of neutrophils or their delayed elimination will disintegrate and release phlogistic cargo that may further contribute to ongoing inflammation, tissue destruction, or autoimmunity [36]. Moreover, activated macrophages produce important cytokines that initiate inflammation, such as IL-1β, IL-6, and TNF-α. Excessive release of these cytokines by activated macrophages can lead to adverse effects such as cell death, severe tissue damage, and disruption of homeostasis [37]. Therefore, how to eliminate or block the function of the inflammatory mediators produced by activated inflammatory cells is a question that needs to be solved urgently in the treatment of many CIDs, such as atherosclerosis, sepsis, and arthritis.

## 3. Anti-Inflammatory Effects of Canthin-6-Ones

The development of CID is closely related to the excessive production of inflammatory mediators and cytokines [38]. Botanical-based immunomodulators are often employed as supportive or adjuvant therapy to overcome the undesired effects of chemotherapeutic agents and to restore health [39]. Canthin-6-one alkaloids display a significant role in the amelioration of many human diseases in animal models due to their various anti-inflammatory properties, and emerging studies have made significant progress in exploring the molecular mechanisms of this anti-inflammatory effect [11,12,13,14,15,16,17]. Canthin-6-one (1) is the parent compound of a large subclass of β-carboline alkaloids with an additional D ring. Since the first canthin-6-one alkaloid was isolated from the Australian plant *Pentaceras australis* by Haynes et al. in 1952, over eighty natural derivatives of this alkaloid family have been isolated from a variety of plant families, such as *Rutaceae*, *Simaroubaceae*, and *Malvaceae* [40]. Recent reports also revealed the existence of canthin-6-one alkaloids in fungi and marine organisms [41,42]. Canthin-6-one (1) was initially synthesized by the classic Bischer-Napieralski method in 1966 with poor efficiency. Subsequently, other methods, such as the Pictet-Spengler reaction, have been described, which have greatly improved the synthetic efficiency and purity of canthin-6-ones [43]. The biosynthesis of canthin-6-ones has been extensively reviewed elsewhere by other researchers [44]. Here, we focus our discussion on the emerging anti-inflammatory effects of canthin-6-one (1) and its derivatives, especially in in vitro and in vivo studies associated with particular CIDs, as well as the related signaling pathways and the regulated pro-inflammatory cytokines and mediators. We summarize the detailed information, including the working doses of the canthin-6-ones, in these studies, which are shown in Table 1. The chemical structures of canthin-6-one and its derivatives described in this study are shown in Figure 1.

### 3.1. Canthin-6-Ones Suppress the Progression of Several CIDs

#### 3.1.1. Inflammatory Bowel Disease (IBD)

IBD is a term to describe a class of diseases that cause inflammatory conditions in the gastrointestinal tract, primarily, Crohn’s disease (CD) and ulcerative colitis (UC) [53,54]. The development of these diseases is associated with many inflammatory factors, including the increase of pro-inflammatory cytokines such as TNF-α, IL-6, and IL-12/23. The inhibition of these cytokines using neutralizing antibodies, such as infliximab and tocilizumab, has been proven to be effective in inducing remission of IBD [55,56,57], although there are also case reports showing that blockade of IL-6 may induce colitis in some patients with autoimmune diseases such as Takayasu arteritis [58,59]. Arunachalam et al. explored the role of canthin-6-one (1) in the management of colitis using a rat model induced by 2,4,6-trinitrobenzene sulfonic acid (TNBS), a hapten that binds to tissue proteins and elicits inflammatory responses in the colon [11]. This model manifests pathological characters close to those found in patients with CD [60,61]. Using this model, the researchers found that canthin-6-one (1) can suppress TNBS-induced colitis based on macroscopic and histologic scoring data and the production of colonic pro-inflammatory mediators, including TNF-α, IL-1β, IL-12p70, and VEGF, while enhancing the level of the anti-inflammatory cytokine IL-10. Further in silico analysis of the molecular targets involved in gut inflammatory signaling revealed that canthin-6-one (1) might bind with p38α and TLR8. Moreover, in a study by Zhang et al. [62], the researchers used a strategy based on ultra-high performance liquid chromatography–tandem mass spectrometry (UPLC-MS/MS) and identified a canthin-6-one alkaloid, 4-methoxy-5-hydroxy-canthin-6-one (6), from rat serum after oral administration of the Liandan Xiaoyan Formula (LXF). This compound could be rapidly absorbed following oral treatment in both dextran sulfate sodium-treated rats with UC conditions and the control animals. However, the role of this compound in UC, its intracellular targets, and inflammatory pathways need to be further studied. The reduction of pro-inflammatory cytokines after exposure to canthin-6-one (1) is consistent with the findings of the in vitro studies by Cho et al. [27] showing that canthin-6-one (1) suppresses LPS-induced production of TNF-α by macrophages via inhibition of NF-κB signaling. The effect of canthin-6-one (1) on suppressing TNF-α is functionally similar to that of neutralizing antibodies mentioned above and many other phytochemicals, such as naringenin [63], and may thus be an alternative drug for the treatment of IBD. Considering that the major source of pro-inflammatory cytokines in intestinal mucosal tissue is myeloid cells such as macrophages and dendritic cells, the effect of canthin-6-one (1) in IBD models may be achieved by suppressing TLR8-mediated NF-κB and MAPK p38 pathways by direct binding, thereby downregulating the production of pro-inflammatory cytokines.

#### 3.1.2. Alzheimer’s Disease (AD)

AD is a disorder that causes the degeneration of brain cells and is the leading cause of dementia, a syndrome characterized by reduced thinking and independence abilities [64,65]. Recent studies have shown that the major pathogenesis of AD is associated with the accumulation of the β-amyloid protein and/or abnormal tau protein, which causes oxidative stress and inflammation; the latter then induces the disturbance of brain cell functions and eventually causes cell death [64]. Guo et al. isolated several canthin-6-one alkaloids from the EtOAc extract of the *Picrasma quassioides* stem in a systematic phytochemistry study. In further functional tests, they found that several canthin-6-ones, including 9-hydroxy-canthin-6-one (10), 4,5-dimethoxy-canthin-6-one (12), 4-methoxy-5-hydroxyl-canthin-6-one (20), 4,5-dimethoxy-10-hydroxy-canthin-6-one (11), and 3-methylcanthin-5,6-dione (3), exhibited protective functions for nerves and thus improve memory and cognitive abilities in mice with AD induced by the amyloid-β peptide. With respect to the mechanisms, they found that these compounds showed potential neuroprotective activity in L-glutamate-stimulated PC12 and Aβ25-35-stimulated SH-SY5Y cell models [12]. This effect may be related to the anti-inflammatory and anti-oxidative activities of the compounds. Several studies have reported the role of 3-methylcanthin-5,6-dione (3) in suppressing NO production [12,47,48,49], while 9-hydroxy-canthin-6-one (10), 4,5-dimethoxy-10-hydroxy-canthin-6-one (11), and 4,5-dimethoxy-canthin-6-one (12) have a role in suppressing the production of pro-inflammatory cytokines [12]. As a neurotransmitter, NO plays a critical role in maintaining the normal function of the neurons, while overproduction of NO may induce nitroxidative stress, which is associated with some pathological changes in AD [66,67]. Moreover, many other natural compounds, such as quercetin and tacrine, can ameliorate AD symptoms via anti-amyloidogenic or anti-cholinesterase activity [68]. However, the available data cannot reveal if canthin-6-ones have similar roles. Based on current findings, canthin-6-one (1) and its numerous derivatives may have an effect on improving particular pathological changes in AD by downregulating pro-inflammatory cytokines and NO. The detailed molecular mechanisms of this effect and the intracellular target molecules of the compounds need to be further investigated. In addition, in comparison to existing drugs for AD treatment which target amyloids, such as aducanumab and gantenerumab [69], the effects of canthin-6-ones in amyloids need to be studied in future investigations. 

#### 3.1.3. Parkinson’s Disease (PD)

PD is a common neurodegenerative disorder characterized by the loss of dopaminergic neurons and the presence of intracytoplasmic-ubiquitinated inclusions [70]. It has been shown that mutations in alpha-synuclein (α-syn) may contribute to the development of this disease [71,72]. A study by Yuan et al. identified canthin-6-one (1) as an α-syn inhibiting compound, which promoted both wild-type and mutant α-syn degradation in a ubiquitin–proteasome-system-dependent manner in PC12 cells [13]. In this study, using CRISPR/Cas9 genome-wide screening technology combined with the GeCKO library and flow cytometry sorting method, proteasome 26S subunit, non-ATPases 1 (PSMD1) was identified as the target gene of canthin-6-one (1). However, the role of PSMD1 in PD needs to be verified in more extensive in vitro and in vivo studies. Considering the protective effects of several canthin-6-ones identified by Guo et al. [12] in their AD study using PC12 cells, as mentioned above, it is worthwhile to verify if PSMD1 has a role in AD, since no study on this topic is available in the literature. Correspondingly, the neuroprotective role of the compounds identified in the Guo et al. study may also play a role in PD; however, in vivo studies are needed to verify this speculation. Thus, the available data support a protective role of canthin-6-one (1) in neuronal cells by targeting PSMD1 and facilitating α-syn degradation, but its potential to be a PD-treating drug needs to be further confirmed. Moreover, the α-syn degrading role of canthin-6-one (1) is similar to the function of irisin [73], a bioactive peptide induced by exercise that is beneficial for health promotion. Further studies are needed to verify if canthin-6-one (1) can act as an alternative medicine to irisin in the treatment of PD. In addition, the anti-inflammatory and anti-oxidant activities of canthin-6-one (1) may also need to be considered when evaluating its effect on PD in animal models, since both factors are considered to be major regulators of PD [74,75]. 

#### 3.1.4. Diabetes Mellitus (DM)

DM is a term to describe a group of diseases that affect the body’s glucose metabolism. Patients with DM usually do not produce enough insulin or do not respond to insulin due to the loss of functional pancreatic β cells, resulting in high blood glucose levels [76]. The role of canthin-6-ones in DM was initially reported by Agrawal et al. Using a streptozotocin–nicotinamide-induced type-II diabetes model in rats, the researchers showed that single oral administration of a partially purified alkaloid basified toluene fraction (PPABTF) isolated from the roots of *Aerva lanata* L. (Amaranthaceae) (AL) can significantly reduce the serum glucose level of rats [14]. In further studies, they identified the main bioactive compound in the PPABTF as 9-hydroxycanthin-6-one (10) using high-performance thin layer chromatography. Although no mechanistical study of 9-hydroxycanthin-6-one (10) was conducted in the Agrawal et al. study in the context of DM, the effect of this compound has been reported by many other studies. For instance, it has a role in promoting the proliferation of neuronal cells (PC12) [12]. If this effect was similarly applicable to pancreatic β cells, it may be an important mechanism to explain the protective role of this compound in DM animals. Furthermore, 9-hydroxycanthin-6-one (10) can suppress the activation of NF-κB and thereby downregulate the production of pro-inflammatory cytokines [26]. As inflammation is also a critical factor that promotes DM progression, the possible inflammatory suppressive effect of 9-hydroxycanthin-6-one (10) is probably the second arm by which this compound protects the host from DM. Therefore, the protective role of 9-hydroxycanthin-6-one (10) in DM animals might be largely dependent on its potential protection of pancreatic cells and the suppression of the inflammatory milieu. However, all these speculations need to be verified by future studies. 

#### 3.1.5. Rheumatoid Arthritis (RA)

RA is an inflammatory condition causing irreversible cartilage and joint damage. Many cell types, such as joint macrophages and fibroblast-like synoviocytes (FLS), play a prominent role in the development of RA [77]. A study by Fan et al. found that a natural canthin-6-one alkaloid isolated from *Picrasma quassioides*, 4-methoxy-5-hydroxycanthin-6-one (6), suppressed the production of NO and the release of TNF-α from LPS-stimulated RAW 264.7 cells. More importantly, oral administration of 4-methoxy-5-hydroxycanthin-6-one (6) ameliorated adjuvant-induced chronic arthritis in rats. The 4-methoxy-5-hydroxycanthin-6-one (6)-treated animals also manifested reduced paw edema after treatment with carrageenan [15]. These observations suggest a possible role of 4-methoxy-5-hydroxycanthin-6-one (6) in RA-associated pathologies by regulating the production of NO and TNF-α by macrophages. It is not surprising to observe that the role of a canthin-6-one compound can suppress the production of TNF-α, since blocking signaling of this cytokine using neutralizing antibodies such as adalimumab and its biosimilars is an effective strategy for treating RA [78]. Furthermore, the inhibition of NO or iNOS using inhibitors is also efficacious for RA treatment, at least, in animal models [79]. Thus, 4-methoxy-5-hydroxycanthin-6-one (6) may be a molecule targeting both TNF-α and NO; however, its efficacy needs to be further determined in more extensive animal models. In addition, whether this compound has effects on the functions of other cell types involved in pathological changes in RA, such as the FLSs, is also worth investigating. Although no study is available regarding the effects of canthin-6-one (1) and its other analogues on this type of diseases, their inhibitory role in the production of inflammatory mediators, as described above, may indicate a potential application of these alkaloids in suppressing some pathological changes in arthritis.

#### 3.1.6. Ulcers

Ulcers are painful sores that may appear anywhere in or on the body, such as in the blood vessels, the lining of the stomach, and the skin. Inflammation triggered by *Helicobacter pylori* (*H. pylori*) bacteria or the persistent application of nonsteroidal anti-inflammatory drugs (NSAIDs) is an important cause of ulcers, especially of those occurring in the gastrointestinal tract [80]. This disorder can also be affected by pathophysiologic events, such as impaired gastric secretions, and environmental factors, such as alcohol consumption and drug ingestion [81,82]. A recent study by De Souza Almeida et al. [83] found that the canthin-6-one alkaloid isolated from methanol-macerated rhizomes of *Simaba ferruginea A. St-Hil*. (*Simaroubaceae*), a herbal plant widely used in traditional medicine for the treatment of gastric ulcers, diarrhea, and fever, may have anti-ulcer activity. Further experiments revealed that pre-treatment of animals with canthin-6-one alkaloids reduced the generation of ulcers induced by ethanol and indomethacin in the gastric tissue in both mice and rats. Moreover, this effect of canthin-6-one (1) was partly mediated by inducing the production of NO and reducing the levels of myeloperoxidase and malondialdehyde. The decline of ulcers induced by canthin-6-one may not be mediated by the downregulation of inflammation in the local environment of the ulcer, since the inflammatory milieu might act as a beneficial factor and promote the healing of ulcers. A good example supporting this speculation is provided by the study by Freitas et al. [84], which showed that blockade of TNF-α delayed the wound healing process of ulcers in a rat model. However, the exact role of inflammation in ulcer healing needs to be further verified. In addition, the exact intracellular targets of canthin-6-ones that mediate its anti-ulcer activity are still unknown, and whether other canthin-6-one derivatives have anti-ulcer activities also needs to be further investigated.

#### 3.1.7. Erectile Dysfunction (ED)

ED is a condition that can be affected by both organic and psychogenic factors. It is also closely associated with inflammatory factors, such as those induced by infections [85]. Previous studies showed that the inhibition of phosphodiesterase-5 (PDE-5) using Tadalafil, a commercially available-carboline medication, is an effective treatment for ED, probably by suppressing PDE-5-mediated inflammatory responses [86,87]. Choonong et al. isolated several canthin-6-one alkaloids from *Eurycoma longifolia Jack* (EL) and *Eurycoma harmandiana Pierre* (EH), including canthin-6-one (1), 9-methoxycanthin-6-one (8), and 9-hydroxycanthin-6-one (10), and found that these canthin-6-ones show potent PDE-5 inhibitory effects [17]. Moreover, it has been shown that ED patients display an increased level of pro-inflammatory cytokines in their blood. Thus, inhibition of these cytokines is one of the common effects through which ED-treating drugs achieves their functions. For instance, sildenafil, a common PDE5 inhibitor used to treat ED, could suppress the blood concentration of many inflammatory mediators, including IL-6 and TNF-α [88]. In agreement with this finding, Sahin et al. [89] showed that blockade of TNF-α could normalize the circulating and cavernosal concentration of many inflammatory mediators, such as TNF-α, CRP, MCP-1, ICAM-1, and testosterone, and proposed that the inhibition of TNF-α may be a promising strategy for treating age-related ED. In addition, the inhibitory role of canthin-6-ones toward NO and iNOS may suggest a potential role of these compounds in improving ED by reducing oxidative stress, which acts as a promoting factor for ED development. Along this line of evidence, Yuan et al. [90] showed that the suppressive effect of liraglutide on ED was, at least, in part achieved by suppressing oxidative stress. Based on these findings, canthin-6-ones may be promising candidates for treating disorders related to male sexual performance by targeting PDE-5, pro-inflammatory cytokines, and NO. Additional research into the mechanism of these compounds for ED treatment will be necessary.

#### 3.1.8. Tumors

The occurrence and progress of tumors are highly associated with chronic inflammation in the tissues [91]. Many canthin-6-ones showing anti-tumor activity may also have potential anti-inflammatory effects (Table 2). A previous study showed that canthin-6-one (1) had a strong anti-proliferation effect on many tumor cell lines, such as human prostate adenocarcinoma PC-3 cells, with low toxicity [92]. Several canthin-6-one derivatives may also have anti-tumor activity. For instance, Yunos et al. found that 9-methoxycanthin-6-one (14) manifests significant anti-tumor effects using a Sulphorhodamine B assay [93]. Their further studies revealed that the effect of this compound is exerted by inducing tumor cell apoptosis. Furthermore, three other canthin-6-ones, including 9-hydroxycanthin-6-one (10), 9-methoxycanthin-6-one-N-oxide (16) and 9-hydroxycanthin-6-one-N-oxide (17), isolated from roots of *Eurycoma longifolia*, were cytotoxic to tumor cells [52]. Jiang et al. showed that four canthin-6-ones isolated from the stem of *Picrasma quassioides* Bennet (Simaroubaceae), including 5-hydroxy-4-methoxycanthin-6-one (15), 4,5-dimethoxycanthin-6-one (12), 8-hydroxycanthin-6-one(18), and 4,5-dimethoxy-10-hydroxycanthin-6-one (11), have a role in reducing cell growth and exhibit significant cytotoxicity to human nasopharyngeal carcinoma (CNE2) cells [47]. In addition, using in silico molecular docking studies, Bultum et al. found that several canthin-6-ones derived from *Brucea antidysentrica*, including canthin-6-one (1), 1-methoxycanthin-6-one (19), 1,11-dimethoxycanthin-6-one (20), and 2-methoxycanthin-6-one (21), are candidates for preventing acute myeloid leukemia (AML) [94]. Along the same line of evidence, Torquato et al. revealed that 10-methoxycanthin-6-one (22) had cytotoxicity toward malignant AML cells by activating necrotic and apoptotic processes, stress-activated MAPKs, and DNA damage pathways. Canthin-6-ones are therefore anticipated to be brand-new weapons in the struggle against leukemia [95]. 

All these observations may indicate a possible application of these canthin-6-ones as chemotherapy drugs for cancers, since drug-induced cytotoxicity is one of the common mechanisms by which chemotherapy drugs kill cancer cells [96,97]. Although few studies have been conducted to investigate their potential effects on inflammation, their roles in cell death may indicate an effect of these compounds on inflammatory responses. During regulated cell death, many caspases can be activated and thereby regulate inflammatory responses. Supporting this line of evidence, studies have shown that exposure to canthin-6-one (1) may induce caspase-8, caspase-9, and caspase-3 activation [98], which are regulators of apoptosis in cancer cells. At the same time, the activation of these caspases, such as caspase 8, may promote activation of the NLRP3 inflammasome, which subsequently induces GSDMD-mediated pyroptosis of the cells [99]. Both inflammasome and pyroptosis are critical regulators of CIDs. This may be one way that canthin-6-ones with anti-tumor activity exert their regulatory roles in CID. Moreover, canthin-6-one (1) can suppress the production of COX2 [29], a critical regulator of autophagy. Studies have shown that COX2 expression is inversely correlated with autophagy [100]. Exposure to canthin-6-one (1) may induce a decline in COX2 levels and a corresponding increase in autophagy. While the effect of autophagy on cancer cells and whether it has a tumor-suppressive or tumor promoting role is dependent on the context and stage of cancer development, enhanced autophagy can suppress inflammatory responses in CID [101]. Thus, regulation of autophagy can be another mechanism through which canthin-6-ones with anti-tumor activity regulate inflammatory processes in CID. However, the role of most of the canthin-6-ones with anti-tumor activity in inflammation has not been studied. Their roles in regulating apoptosis, necrosis, and pyroptosis, as well as their potential role in regulating the inflammasome and autophagy may indicate a possible effect of these compounds in regulating inflammatory diseases.


**Table 2 molecules-28-03381-t002:** Canthin-6-ones exert anti-tumor activity via various mechanisms.

Compound No.	Compound Name	Source	Study Type	Effective Dose/IC50	Reference
(10)	9-Hydroxycanthin-6-one	*Eurycoma longifolia*	In vitro	Suppressed the proliferation of a melanoma cell line (5.4 μM)	[52]
(11)	4,5-Dimethoxy-10-hydroxycanthin-6-one	*Picrasma quassioides* (D. Don) Benn	In vitro	Suppressed the proliferation of CNE2 (11.6 ± 2.48 μM) and Bel-7402 (118.91 ± 67.42μM) cell lines	[47]
(12)	4,5-Dimethoxy- canthin-6-one	*Picrasma quassioides* Benn (*Simarobaceae*)	In vitro	Suppressed the proliferation of CNE2 (9.86 ± 1.49 μM) and Bel-7402 (32.27 ± 9.74 μM) cell lines	[47]
(14)	9-Methoxycanthin-6-one	*Eurycoma longifolia*	In vitro	Suppressed the proliferation of A2780 (4.04 ± 0.36 μM), SKOV-3 (5.80 ± 0.40 μM), MCF-7 (15.09 ± 0.99 μM), HT-29 (3.79 ± 0.069 μM), A375 (5.71 ± 0.20 μM), and HeLa (4.30 ± 0.27 μM) cell lines	[29,93]
(15)	6-Hydroxy-4-methoxy canthin-6-one	*Picrasma quassioides* BENN	In vitro	Suppressed the proliferation of CT26.WT, K-562, SGC-7901, Hep G2, and A-549 cell lines (>50 μM)	[47,102]
(16)	9-Methoxycanthin-6-one-N-oxide	*Eurycoma longifolia*	In vitro	Suppressed the proliferation of a melanoma cell line (6.5 μM)	[52,93]
(17)	9-Hydroxycanthin-6-one-N-oxide	*Eurycoma longifolia*	In vitro	Suppressed the proliferation of a melanoma cell line (7.0 μM)	[52]
(18)	8-Hydroxycanthin-6-one	*Picrasma quassioides* (D. Don) Benn	In vitro	Suppressed the proliferation of CNE2(13.43 ± 2.29 μM) and Bel-7402(39.27 ± 9.72 μM) cell lines	[47]
(19)	1-Methoxy-canthin-6-one	*Ailanthus altissima Swingle*	In vitro	Suppressed the proliferation of thyroid carcinoma and hepatocellular carcinoma cell lines (40 μM).	[103]
(20)	1,11-Dimethoxycanthin-6-one	*Brucea antidysentrica*	In silico	Showed consecutive binding affinity with mainly hydrophobic interaction (−11.0 kcal/mol)	[94]
(21)	2-Methoxycanthin-6-one	*Brucea antidysentrica*	In silico	Showed consecutive binding affinity with mainly hydrophobic interaction (−11.9 kcal/mol)	[94]
(22)	10-Methoxycanthin-6-one	Chemical synthesis	In vitro	Suppressed the proliferation of Kasumi-1 (80μM) and KG-1 (36μM)	[95]

### 3.2. Mechanism of Action and Pathways for Canthin-6-Ones

The above studies provided evidence that canthin-6-ones might have a potential application in the treatment of many kinds of inflammatory diseases, such as CIDs. Recent reports also reveal the intracellular signaling pathways that canthin-6-ones might target to thus exert their anti-inflammatory functions. In this section, we provide an overview of these signaling pathways and discuss their potential contributions to the anti-inflammatory role of canthin-6-ones (Figure 2).

#### 3.2.1. NF-κB Pathway

The transcription factor NF-κB is a central regulator of inflammation and immune responses. It can be activated by a variety of environmental cues, including inflammatory cytokines, pattern recognition receptor ligands, and endogenous danger signals [104]. Overactivation of NF-κB can lead to serious diseases such as CIDs, autoimmune diseases, and cancers. Recent studies have shown that inhibition of NF-κB is an effective approach for the management of numerous diseases [105]. A recent study by Yue et al. showed that canthin-6-one (1) pretreatment could significantly inhibit LPS-induced activation of molecules in the NF-κB pathway, such as the phosphorylation of IκBα, IKKα/β, and NF-κB p65 in astrocytes [25]. Thus, canthin-6-one (1) may provide significant benefits by suppressing astrogliosis via the regulation of NF-κB signaling in neurodegenerative disorders. Moreover, Tran et al. reported that canthin-6-one derivatives may also work as NF-κB inhibitors. They showed that IC50 values of 9-methoxy-canthin-6-one (8) and 9-hydroxycanthin-6-one (10) for NF-κB inhibition were 3.8 μM and 7.4 μM, respectively, which were higher than that of the standard drug, valuliolide, with an IC50 of 1.5µM [26]. Moreover, Cho et al. demonstrated that canthin-6-one (1) without a hydroxyl group in the D ring could downregulate NF-κB activity in LPS-stimulated macrophages [27].

Additionally, a study by Yue et al. [25] showed that canthin-6-one (1) may affect the activation of the NLRP3 inflammasome, a cytoplasmic protein complex composed of a sensor (NLRP3), an adapter (ASC), and an effector (caspase-1) [106]. Assembly of the NLRP3 inflammasome occurs when cells are disturbed, and this assembly leads to the activation of caspase-1, which promotes the maturation and release of inflammatory cytokines, including IL-1β and IL-18. Activation of caspase-1 induces inflammatory cell death, termed pyroptosis [106]. Aberrant activation of the NLRP3 inflammasome may drive chronic inflammation in vivo and regulate the pathogenesis of many inflammation-related diseases [55,56,57]. Yue et al. [25] showed that canthin-6-one (1) pre-treatment could significantly suppress LPS-induced expression of NLRP3 and pro-caspase-1 in astrocytes. Regarding the mechanisms, they showed that canthin-6-one (1) may inhibit the activation of NF-κB and STAT3 pathways. However, whether canthin-6-one derivatives affect the activation of the NLRP3 inflammasome needs to be further determined.

#### 3.2.2. MAPK Pathway

MAPKs are a group of serine/threonine protein kinases that play a role in regulating cell growth and death in response to a range of stimuli, such as osmotic stress, and inflammatory cytokines [107]. The three members of MAPKs, including extracellular signal-regulated kinase (ERK), p38, and c-Jun NH(2)-terminal kinase (JNK), can be activated by many cellular signals and act as ligands of cytokine receptors and stress sensors in inflammatory responses [108,109,110]. Regarding the effects of canthin-6-ones on MAPKs, in a study by Yue et al. [25], the authors found that canthin-6-one (1) treatment dramatically suppressed the phosphorylation of ERK, p38, and JNK. However, whether the derivatives of canthin-6-one (1) have a role in MAPK suppression needs to be determined. Moreover, it is worthwhile evaluating the significance of MAPK inhibition by canthin-6-one (1) in inflammatory diseases in future studies. 

#### 3.2.3. JAK/STAT Pathway

The JAK/STAT pathway is a signaling cascade that transduces signals from many cytokine receptors and is involved in the pathogenesis of a wide range of CIDs and cancers. JAK inhibitors are effective in treating many CIDs, such as rheumatoid arthritis and psoriasis [111]. In investigating the effects of canthin-6-one (1) on MAPKs, Yue et al.’s study also showed that this compound has a role in suppressing the phosphorylation of STAT3 [25], indicating an impact of canthin-6-one (1) in regulating the JAK/STAT pathway. Whether the suppression of this pathway by canthin-6-one (1) can be applied to treating an inflammatory disease need to be further determined.

#### 3.2.4. PI3K–AKT Pathway

The PI3K–AKT pathway is an intracellular mechanism that transduces signals to regulate several physiological processes. The activation of this pathway can be induced by several events, which primarily include the binding of ligands such as growth factors. Previous studies have shown that the PI3K–AKT pathway inhibits LPS-induced inflammatory mediators in monocytes, microglia cells, and endothelial cells [112,113]. As mentioned above, Cho et al. [27] provided evidence that the anti-inflammatory effect of canthin-6-one (1) without a hydroxyl group in the D ring could be partially attributed to its role in suppressing the AKT pathway in LPS-stimulated macrophages. Meanwhile, the study by Yue et al. [25] found that treatment with canthin-6-one (1) can suppress the phosphorylation of AKT and increase endothelial nitric oxide synthase (eNOS) expression, indicating an inhibitory role of canthin-6-one (1) in the PI3K–AKT pathway. This impact is inconsistent with the anti-inflammatory role of the compound. Thus, the negative regulation of the PI3K–AKT pathway may be a concomitant event that occurs simultaneously with the inhibition of other inflammatory pathways induced by canthin-6-one (1). Its physiological significance needs to be further investigated.

### 3.3. Major Inflammatory Mediators Regulated by Canthin-6-Ones

As described above, canthin-6-ones affects many signaling pathways, and the subsequent activation or inactivation of these signaling pathways might then modulate the production and many cytokines and inflammatory mediators, thus regulating the progression of inflammatory responses. In this section, we summarize the major inflammation-associated mediators and the regulators that could be modulated by canthin-6-ones (Figure 3).

#### 3.3.1. Pro-Inflammatory Cytokines

Pro-inflammatory cytokines are mediators predominantly produced by antigen-presenting cells that play a key role in upregulating inflammatory reactions. Studies have shown that several major pro-inflammatory cytokines, such as IL-1β, IL-6, and TNF-α, contribute to the host’s response against infections and also play a role in the development of many chronic diseases. Some of the canthin-6-ones may affect the production of pro-inflammatory cytokines. For instance, Zhao et al. showed that β-carboline alkaloids can inhibit the release of TNF-α and IL-6 in lipopolysaccharide (LPS)-activated RAW 264.7 macrophage cells [28]. Yue et al. found that canthin-6-one (1) from *Picrasma quassioides* (D.Don) Benn can suppress LPS-induced astrocyte activation and the associated production of pro-inflammatory cytokines, including that of TNF-α, IL-6, and IL-1β [25]. Fan et al. also revealed that 4-methoxy-5-hydroxy-canthin-6-one (6) can significantly suppress the production of TNF-α [15]. Moreover, Liu et al. [29] explored the possible mechanism of the inhibitory effects of 9-methoxycanthin-6-one (8) on the LPS-induced production of pro-inflammatory cytokines such as TNF-α, IL-1β, and IL-6 in macrophages from BALB/c mice and found that this compound significantly inhibited inducible nitric oxide synthase (iNOS) and COX-2 in LPS-stimulated macrophages, a consequence that was not attributable to cytotoxicity, as assessed by the expression of the housekeeping gene GAPDH. Thus, canthin-6-one (1) and its several derivatives suppress the production of pro-inflammatory cytokines.

#### 3.3.2. NO

NO is a biological messenger molecule and neurotransmitter synthesized by NO synthases such as iNOS in multiple cells [114]. The NO synthase iNOS acts as a key mediator of immune responses and inflammation [115], and its expression can be induced by extracellular stimulation such as LPS [116]. The dysregulation of iNOS has been linked to several diseases. It has been shown that β-carboline alkaloids, including quassidine F, 6-methoxy-3-vinyl-β-carboline, and 6,12-dimethoxy3-vinyl-β-carboline, have a role in suppressing the production of NO in LPS-activated RAW 264.7 macrophage cells. Moreover, 3-methylcanthin-5,6-dione (3), a derivative of β-carboline that has antioxidant activity, inhibited LPS-stimulated NO production in RAW264.7 cells [48,49]. The impact of these alkaloids is largely mediated by regulating iNOS since downregulation of iNOS protein expression or inhibition of the iNOS enzymatic activity blocks the inhibitory role of these alkaloids on NO production [28]. Thus, blockade of iNOS has been suggested to be the theoretical basis for using β-carboline analogs in the treatment of CIDs.

As a subtype of β-carboline alkaloids, it is reasonable to assume that canthin-6-ones may also have an inhibitory effect on NO production. For example, Fan et al. evaluated the anti-inflammatory effect of a canthin-6-one derivative, 4-methoxy-5-hydroxycanthin-6-one (6), and found that this compound significantly inhibited LPS-induced NO release while downregulating iNOS expression in RAW264.7 cells [15]. Canthin-6-one (1) and 9-methoxy-canthin-6-one (8) isolated from Simaroubaceae plants have been shown to suppress NO production in LPS-activated RAW 264.7 macrophages. Another study by Yue et al. showed that canthin-6-one (1) could suppress the production of NO from astrocytes and thus play a neuroprotective effect [25]. Moreover, Liu et al. [29] found that canthin-6-one (1) displayed significant inhibitory activity against NO production and the expression of iNOS in a dose-dependent manner in LPS-activated RAW264.7 macrophages. In addition, Kim et al. [117] identified three new canthin-6-one type alkaloids from the stem barks of *Ailanthus altissima*, including canthin-6-one-1-O-β-D-apiofuranosyl-(1→2)-β-D-glucopyranoside, canthin-6-one-1-O-(6-O-(3-hydroxy-3-methylglutaryl))-β-D-glucopyranoside, and canthin-6-one-1-O-(2-β-D-apiofuranosyl-6-O-(3-hydroxy-3-methylglutaryl))-β-D-glucopyranoside, and found that they all suppress the production of LPS-induced NO in RAW 264.7 cells. Zhang et al. isolated two new canthin-6-one alkaloids, 4,9-dimethoxy-5-hydroxycanthin-6-one and 9-methoxy-(R/S)-5-(1-hydroxyethyl)-canthin-6-one, from the roots of Thailand *Eurycoma longifolia* Jack and found that both compounds could inhibit NO release from RAW264.7 cells [118].

#### 3.3.3. PGE2

Prostaglandins (PGs) are the major lipid mediators in animals and are biosynthesized from arachidonic acid by cyclooxygenases as rate-limiting enzymes. PGE2 is the most abundant PG in various tissues and performs a variety of physiological and pathological functions [119]. PGE2 plays an important role in cell growth and causes inflammatory symptoms. Two cyclooxygenases (COX-1 and COX-2) are responsible for catalyzing the initial steps of arachidonic acid metabolism and prostaglandin synthesis and act as the primary agent of inflammation in mammalian cells. COX-2 is mainly expressed in inflammatory cells and is significantly upregulated in chronic and acute inflammation, becoming a key target for many drug inhibitors [120].

A good example of the impact of canthin-6-ones on PGE2 production was provided by the study by Liu et al. In this study, the researchers investigated the possible anti-inflammatory effects of several β-carboline-type alkaloids and canthin-6-one-type alkaloids isolated from many Simaroubaceae plants [29]. They found that the canthin-6-one-type alkaloids, including canthin-6-one (1) and 9-methoxy-canthin-6-one (8), can inhibit the production of PGE2 and COX-2 in LPS-activated RAW 264.7 cells. Interestingly, these two compounds can also inhibit the production of iNOS and NO. In contrast, the β-carboline-type alkaloids, including benzalharman, kumujian, 1-ethyl-1,2,3,4-tetrahydro-β-carboline-3-carboxylic acid, and 1-acetophenone-1,2,3,4-tetrahydro-β-carboline-3-carboxylic acid, inhibit iNOS and NO production but do not affect the production of PGE2 and COX-2. Based on these findings, canthin-6-one-type alkaloids are a functionally distinct subtype of β-carboline-type alkaloids with unique anti-inflammatory activities. 

## 4. Structure–Activity Relationships of Canthin-6-Ones

Many previous studies have demonstrated that the structure of a compound can significantly affect its biological functions [121]. Using IC50 values in LPS-treated RAW 264.7 cells, Kim et al. assessed the relative inhibitory effects of four canthin-6-one-type compounds, namely, canthin-6-one (1), 10-hydroxycanthin-6-one(2), 9-hydroxycanthin-6-one(10), and (R)5-(1-hydroxyethyl)-canthin-6-one (7), on NO production [45]. They found that 9-hydroxycanthin-6-one (10) (IC50 = 7.73 µM) exhibited higher inhibitory activity than canthin-6-one (IC50 = 9.09 µM), whose inhibitory activity was higher than that of (R)5-(1-hydroxyethyl)-canthin-6-one (7) (IC50 = 15.09 μM), suggesting that hydroxylation at site 9 may increase, while a 1-hydroxyl group in the C-5 position may reduce, the inhibitory activity of canthin-6-one (1) on NO production. These findings suggest that the presence of a hydroxyl group at different sites of canthin-6-one (1) is highly likely to change its inhibitory effect on NO production. However, whether other structural groups affect the activities of canthin-6-one (1) needs to be further determined.

Regarding the relationship between the structure of canthin-6-ones and their bactericidal and fungicidal activities, Zhao et al. [46] proposed that the conjugated unsaturated bonds of rings C and D are non-essential for the antifungal activities of canthin-6-ones, while this modification is essential for the antibacterial effects of the compounds. In contrast, the aliphatic ester modifications may improve the antifungal activities of canthin-6-ones, whereas the introduction of aromatic esters inhibits their antibacterial effects.

## 5. Limitations and Possible Solutions

As described above, canthin-6-one (1) and its numerous derivatives show significant ability to suppress the activation of various inflammatory pathways, inhibit infections, and limit tumor cell proliferation. These properties of canthin-6-ones may benefit the host, whereas several studies using cancer cell lines have proposed the potential cytotoxicity of these compounds [122]. For instance, two canthin-6-ones isolated from the roots of *Eurycoma longifolia* displayed significant cytotoxicity against human lung cancer (A-549) and human breast cancer (MCF7) cell lines [123]. Moreover, Jiang et al. [47] showed that several canthin-6-one alkaloids isolated from the stem of *Picrasma quassioides* Bennet (Simaroubaceae) exhibited significant cytotoxic activity against human CNE2 cells. This character of canthin-6-ones may contribute to their anti-tumor activity, which is especially important for the development of chemotherapy drugs. However, the cytotoxicity of these canthin-6-ones may limit their applications in the treatment of CIDs and the management of infections. A possible approach that may reduce the cytotoxicity of these canthin-6-one alkaloids is to reduce the working concentrations of the compounds such that they still exhibit significant biological activities while having negligible killing activity toward normal host cells. Furthermore, the drug sensitivity of a particular cell line is also likely to affect the outcome. Thus, extensive assessment of the efficiency and safety of a compound in numerous cell lines and animal models is a must before its clinical application.

Another point that may limit the further investigation of canthin-6-ones is the fact that, until now, few cellular targets that directly bind to the compounds have been identified. The compound-binding partners may play a key role in the regulation of canthin-6-one-induced signaling pathways. The application of predicting tools for compound-binding molecules may benefit further functional studies of the compounds. Moreover, further transcriptomic or proteomic studies are needed to examine their expression patterns in particular cell systems, which may determine the possible involvement of the targets in a biological process. Further bioinformatic analysis of the predicted proteins may also be helpful for making connections from the predicted targets to particular pathways. In addition, more studies are needed to confirm the binding of the predicted molecules with canthin-6-ones and to investigate the roles and associated mechanisms of these molecules in the anti-inflammatory effects of canthin-6-ones using various biological methods such as via a gene knock-down or knock-out approach.

Based on the available functional studies, the inhibitory effects of canthin-6-ones on pro-inflammatory mediators such as TNF-α, NO, and PGE2 are quite similar to some other phytochemicals (such as naringenin, a flavonoid that has anti-colitis activity and many other biological activities [63]), although the affected signaling pathways may be different. Thus, the functional similarity of the novel drugs may provide new options for the treatment of a disease to overcome drug resistance induced by long-term usage of a particular chemical [124]. Moreover, most available drugs for the treatment of CIDs such as IBD may cause severe adverse reactions, which limit the long-term application of a drug despite its efficacy to a disease [125]. Drugs that display similar functions but variable adverse effects in different tissues may offer alternative strategies for disease treatment. Thus, further studies are required to clarify the drug toxicities or adverse effects in extensive in vivo experiments and thereby achieve the possibility of reasonable drug selection.

The unique properties of canthin-6-ones compared with other phytochemicals may mainly be reflected in their binding partners and the signaling pathways that are affected by the drug. However, the current studies of canthin-6-ones cannot tell the difference between chemicals at this level. The uniqueness of a chemical may be largely dependent on its structure. As mentioned above, studies using molecular docking and proteome-based strategies [126] may identify the binding partners of canthin-6-ones and the corresponding signaling pathways that are directly regulated by these compounds in different disease environments. On this basis, studies of disease pathogenesis have provided numerous target molecules for disease treatment. A subsequent functional study that combines the binding partners of canthin-6-ones and the disease targets could lead to the identification of the key regulators through which canthin-6-ones exert their anti-inflammatory roles in a disease.

## 6. Conclusions and Perspectives

Recent studies revealed that canthin-6-ones extracted from a variety of species exhibit anti-inflammatory effects [25,26,27,29,46]. These compounds inhibit the excessive release of pro-inflammatory cytokines and mediators (e.g., IL-1β, IL-6, TNF-α, NO, and PGE2) in many cell types, such as macrophages and astrocytes, through regulating the activation of several signaling cascades, such as the NLRP3 inflammasome, NF-κB, and PI3K-Akt pathways. Several in vivo studies have demonstrated that the administration of several canthin-6-ones has a role in suppressing the progression of many entities of CID, such as Crohn’s disease and Parkinson’s disease. It is difficult to evaluate the potential of canthin-6-ones for further development based on the currently available evidence. A further investigation aiming to assess the relationship between chemical structures and their anti-inflammatory activities using multiple species of animal models may largely improve the treatment efficacy and specificity of canthin-6-ones. However, several questions associated with future studies of canthin-6-one remain to be answered: (1) What are the intracellular targets or binding partners of canthin-6-ones that facilitate their signaling transduction in regulating the production of inflammatory mediators? (2) What are the downstream metabolites of canthin-6-ones and their roles in regulating cell function? (3) How do canthin-6-ones affect other signaling pathways associated with inflammation, such as the Hippo and Wnt pathways? (4) What is the exact relationship between canthin-6-one (1) structures and their anti-inflammatory activity or other biological activities? (5) How can the cytotoxicity of canthin-6-ones in treating CIDs be reduced? The answers to these questions may further improve our understanding of the roles and mechanisms of canthin-6-one (1)-like compounds in regulating the production of inflammatory mediators and their subsequent applications in disease treatment. Future studies in the field of metabonomics and proteomics using several new technologies may accelerate the identification of new regulators in canthin-6-ones-associated signaling pathways, which may further promote the development of new targeted therapies for CIDs.

## Figures and Tables

**Figure 1 molecules-28-03381-f001:**
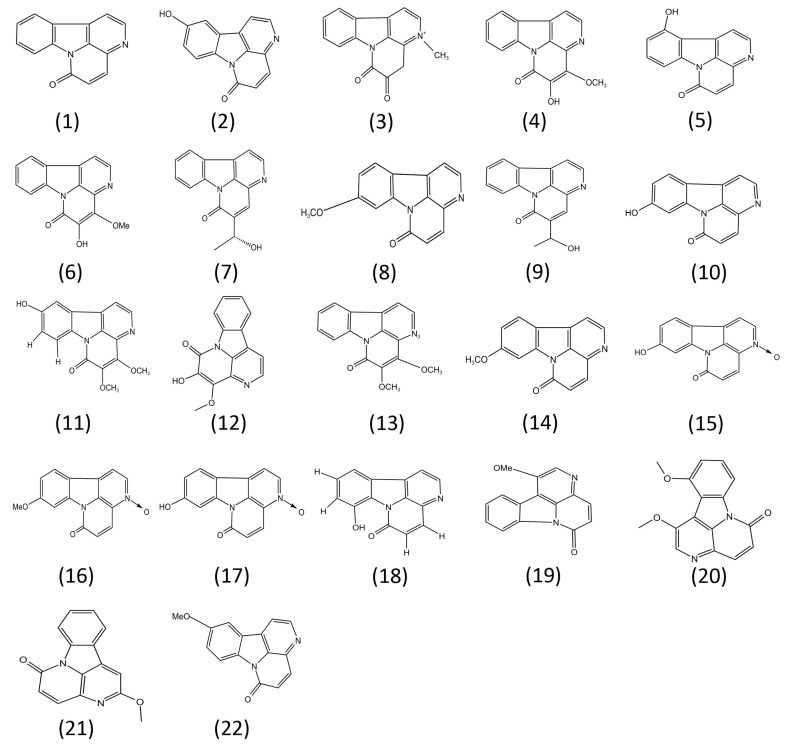
Chemical structures of canthin-6-one and its derivatives. Compound numbers correspond to the compound names in Table 1 and Table 2.

**Figure 2 molecules-28-03381-f002:**
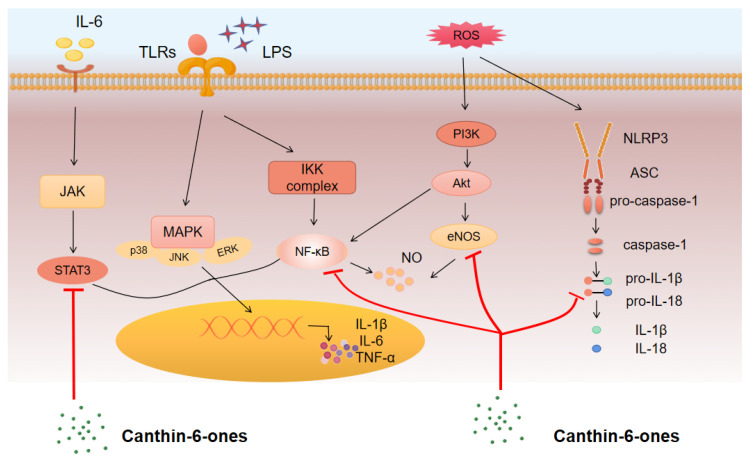
Canthin-6-ones regulate multiple signaling pathways involved in inflammation. Canthin-6-one and its many derivatives suppress the production of pro-inflammatory cytokines by targeting the activation of various signaling pathways, such as the NF-κB, JAK/STAT, and MAPK pathways. The inhibitory role of canthin-6-one in the PI3K–AKT pathway may be a concomitant event that occurs simultaneously with the inhibition of other inflammatory pathways induced by canthin-6-one. The NLRP3 inflammasome was also recently shown to be a target of canthin-6-one, which suppresses both the production and the activation of NLRP3 inflammasome components. Canthin-6-one suppresses the phosphorylation of Rac1, a key signaling molecule of the Rho–Rock pathway in inflammation and endothelial stabilization, indicating a role of this compound in endothelial repair.

**Figure 3 molecules-28-03381-f003:**
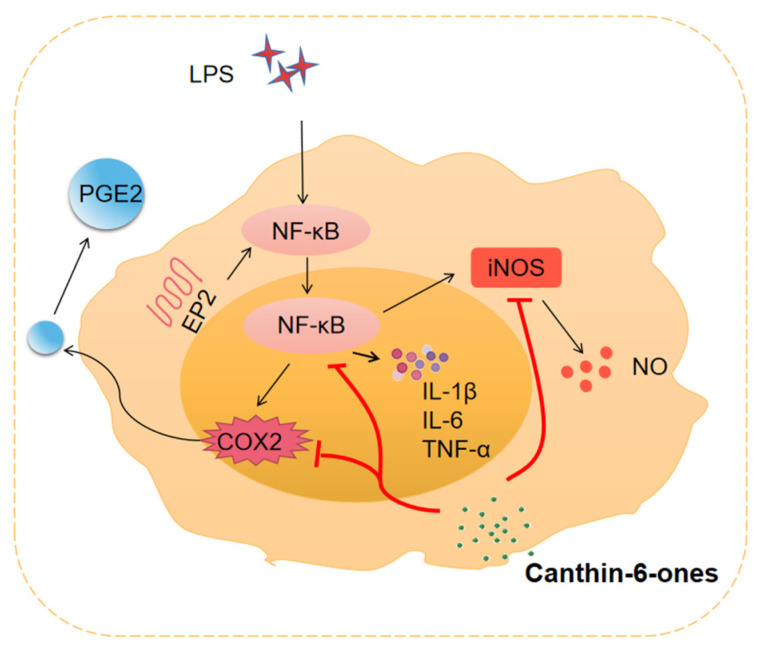
Target molecules of canthin-6-ones. Canthin-6-ones inhibit the production of pro-inflammatory cytokines, such as IL-1β, IL-6, and TNF-α, whose production can be induced by a wide variety of intracellular or extracellular stimuli, such as pathogen-derived LPS. Moreover, studies have shown that canthin-6-ones can suppress the production of NO and its regulator, iNOS. In addition, the production of PGE2 and its regulator, COX2, can be suppressed by canthin-6-ones. Activation of NF-κB mediates the production of NO, iNOS, COX2, and PGE2, which recruit many inflammatory cells, such as macrophages, to the inflammatory sites and promote chronic inflammation.

**Table 1 molecules-28-03381-t001:** Canthin-6-ones exert anti-inflammatory effects by regulating various signaling pathways.

Compound No.	Compound Name	Source	Study Type	Effects on Disease Models	Effects on Inflammatory Pathways	Reference
(1)	Canthin-6-one	Several plants, including the initial source, *Pentaceras australis*	In vivo /In vitro	Suppresses colitis model in Wistar rats by oral gavage (1, 5, and 25 mg/kg)	Suppresses the expression of NO and COX2 (12.5, 25, and 50 μM), PDE-5 (4.31 ± 0.52 μM), NF-κB, MAPK, STAT3, PI3K/Akt, and the NLRP3 inflammasome (6.5, 12.5, and 25 μM)	[15,17,25,27,29,40,45]
(2)	10-Hydroxycanthin-6-one	Chemical synthesis	In vitro	Unknown	Suppresses the expression of NO (5.92 ± 0.9~15.09 ± 1.8 μM)	[45,46]
(3)	3-Methylcanthin-5,6-dione	*Picrasma quassioides* (D. Don) Benn	In vivo /In vitro	Suppresses Alzheimer’s disease model in ICR mice by oral administration (25, 50, 100 mg/kg)	Suppresses the expression of NO (3~30 μM)	[12,47,48,49]
(4)	1:4-Methoxy-5-hyroxycanthin-6-one	*Picrasma quassiodes* (D. Don) Benn.	In vivo	Suppresses artery hypertension disease model in spontaneously hypertensive rats by oral gavage. Suppresses the expression of NO (50, 100, 200 mg/kg)	Unknown	[50]
(5)	11-Hydroxycanthin-6-one	*Brucea mollis var. tonkinensis*	In vitro	Unknown	Suppresses the expression of NO (5.92 ± 0.9~15.09 ± 1.8 μM)	[45,51]
(6)	4-Methoxy-5hydroxy-canthin-6-one	*Picrasma quassioides*	In vivo/In vitro	Suppresses chronic arthritis model in male Sprague–Dawley (SD) rats by oral administration (3, 9, 27 mg/kg)	Suppresses the expression of NO (10, 30, and 100 μM)	[15]
(7)	(R)-5-(1-hydroxyethyl)-canthin-6-one	*Ailanthus altissima Swingle*	In vitro	Unknown	Suppresses the expression of NO (5.92 ± 0.9~15.09 ± 1.8 μM)	[45]
(8)	9-Methoxy-canthin-6-one	*Simaroubaceae* plants	In vitro	Unknown	Suppresses the expression of NO and COX2 (12.5, 25, and 50 μM) or the activity of PDE-5 (3.30 ± 1.03 μM)	[15,17,26,29]
(9)	5-(1-Hydroxyethyl)- 6-canthin-6-one	*Ailanthus altissima*	In vitro	Unknown	Suppresses NF-κB pathway (15 μM)	[27]
(10)	9-Hydroxycanthin-6-one	*Eurycoma longifolia*	In vivo /In vitro	Suppresses Alzheimer disease model in mice by oral administration (25, 50, 100 mg/kg)	Suppresses NF-κB pathway (3.8 μM) and the activity of PDE-5 (4.66 ± 1.13 μM)	[12,17,26,45,52]
(11)	4,5-Dimethoxy-10-hydroxycanthin-6-one	*Picrasma quassioides* (D. Don) Benn	In vivo/In vitro	Suppresses Alzheimer disease model in mice by oral administration (25, 50, 100 mg/kg)	Suppresses the expression of pro-inflammatory cytokines, including IL-1β, IL-6, and TNF-α (25 μM, 50 μM, and 100 μM)	[12]
(12)	4,5-Dimethoxy- canthin-6-one	*Picrasma quassioides* BENNET	In vivo/In vitro	Suppresses Alzheimer disease model in mice by oral administration (25, 50, 100 mg/kg)	Suppresses the expression of pro-inflammatory cytokines, including IL-1β, IL-6, and TNF-α (25 μM, 50 μM, and 100 μM)	[12]
(13)	4-Methoxy-5-hydroxyl- 5-canthin-6-one	*Picrasma quassioides* BENNET	In vivo/In vitro	Suppresses Alzheimer disease model in mice by oral administration (25, 50, 100 mg/kg)	Suppresses the expression of pro-inflammatory cytokines, including IL-1β, IL-6, and TNF-α (25 μM, 50 μM, and 100 μM)	[12]

## Data Availability

The original contributions presented in this study are included in the article. Further inquiries can be directed to the corresponding author.

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
