# Peer review of "Canthin-6-Ones: Potential Drugs for Chronic Inflammatory Diseases by Targeting Multiple Inflammatory Mediators"

_molecules, 2023, doi:10.3390/molecules28083381_

Round 1
Reviewer 1 Report
The manuscript titled 'Canthin-6-ones: Potential Drugs for Chronic Inflammatory Dis- 2 eases by Targeting Multiple Inflammatory Mediators' provides an overview on several diseases, inflammatory mediators, and signaling pathways that have been shown to be affected by canthin-6-ones.
This work is purely descriptive without in-depth discussion relating the different results of the different research papers, which can draw a general scheme of the mechanisms of action of these molecules.
Reviewer 2 Report
The manuscript by Zhang et al addressed the role of Canthin-6-ones as potential drugs for chronical inflammatory diseases. Moreover, they describe the target pathways for this molecule and also emphasize the immunological mediators that could be target for canthin-6 one or its derivates. This manuscript is well organized and supported by references. Also, its reading is fluid. My main concern is following:
In the point 3.1 the authors addressed diseases characterized by inflammation, however in some diseases the focus in inflammation is missing. Also, the role of canthin-6-one or its derivatives in regulate the inflammatory process in this diseases should be emphasized. In this same line, the table 2 is focused on antitumoral effect of canthin-6-ones, however the connection with inflammation is roughly addressed. Additional information to link cancer with inflammation process could be helpful to highlight the effect of this molecule in this disease. For example, numerous reports have pointed that defects in autophagy could be the common denominator of inflammatory diseases, could canthin-6-ones regulate autophagy?
Round 2
Reviewer 1 Report
The authors have made significant improvements to the manuscript titled "Canthin-6-ones: Potential Drugs for Chronic Inflammatory Diseases by Targeting Multiple Inflammatory Mediators" and the discussion section in particular has been enhanced with valuable insights.
The only minor point to be adressed is to add references after statments for which the authors forgot to add them:
examples:
Line 191: Please add a reference.
Lines 271-273. Please add a reference. You may add https://doi.org/10.1016/j.biopha.2022.113126
Lines 295-301. Please add (a) reference(s)
Please carefully review the entire manuscript to ensure that all references are properly inserted. It is important that all sources cited in the text are correctly referenced and that the reference list is complete and accurate.
